# Indocyanine Green (ICG) and Colorectal Surgery: A Literature Review on Qualitative and Quantitative Methods of Usage

**DOI:** 10.3390/medicina59091530

**Published:** 2023-08-24

**Authors:** Laurentiu Simion, Sinziana Ionescu, Elena Chitoran, Vlad Rotaru, Ciprian Cirimbei, Octavia-Luciana Madge, Alin Codrut Nicolescu, Bogdan Tanase, Irinel-Gabriel Dicu-Andreescu, Denisa Mihaela Dinu, Dan Cristian Luca, Dana Lucia Stanculeanu, Adelina Silvana Gheorghe, Daniela Zob, Marian Marincas

**Affiliations:** 1Ist Clinic of General Surgery and Surgical Oncology, Bucharest Oncology Institute, 022328 Bucharest, Romania; dr.simion.laurentiu@gmail.com (L.S.);; 2“Carol Davila” University of Medicine and Pharmacy, 050474 Bucharest, Romania; 3Ph.D. Studies, “Carol Davila” University of Medicine and Pharmacy, 050474 Bucharest, Romania; 4University of Bucharest, 030018 Bucharest, Romania; 5Emergency Hospital “Prof. Dr. Agrippa Ionescu”, 011356 Bucharest, Romania; 6Clinic of Thoracic Surgery, Bucharest Oncology Institute, 022328 Bucharest, Romania; 7Surgery Clinic, Bucharest Emergency University Hospital, 050098 Bucharest, Romania; 8Oncology Clinic, “Prof. Dr. Al. Trestioreanu” Bucharest Oncology Institute, 022328 Bucharest, Romania; 9Ph.D. Studies in Oncology, “Carol Davila” University of Medicine and Pharmacy, 050474 Bucharest, Romania; 10Oncology Department, “Prof. Dr. Al. Trestioreanu” Bucharest Oncology Institute, 022328 Bucharest, Romania

**Keywords:** colorectal, fluorescence, ICG, ICG-NIR, colorectal surgery, intraoperative staining, q-ICG

## Abstract

*Background*: Due to its many benefits, indocyanine green (ICG) has gained progressive popularity in operating rooms (ORs) globally. This literature review examines its qualitative and quantitative usage in surgical treatment. *Method*: Relevant terms were searched in five international databases (1. Pubmed, 2. Sciencedirect, 3. Scopus, 4. Oxfordjournals, 5. Reaxys) for a comprehensive literature review. The main benefits of using ICG in colorectal surgery are: intraoperative fluorescence angiography; fluorescence-guided lymph node involvement detection and the sentinel technique; the fluorescent emphasis of a minute liver tumour, counting just 200 tumour cells; facilitation of fistula diagnosis; and tumour tattooing. This methodology can also be used with quantitative characteristics such as maximum intensity, relative maximum intensity, and in-flow parameters such as time-to-peak, slope, and t1/2max. This article concludes that fluorescence surgery with ICG and near-infrared (NIR) light is a relatively new technology that improves anatomical and functional information, allowing more comprehensive and safer tumour removal and the preservation of important structures.

## 1. Introduction

Adjunctive techniques to better identify, visualize, and resect a specific structure/tissue during an intervention have evolved alongside general surgical techniques, eventually leading to robotic surgery. Fluorescence illuminates vessels, lymph nodes, and malignancies. NIR-ICG (near infrared light plus indocyanine green) enhances the visibility of organs such as the ureter, reducing the likelihood of lesion formation. The current medical literature discusses quantitative ICG utilisation using: time-to-fluorescence, alongside contrast-to-background ratio, as well as the measurement of pixel intensity, and numerical classification score. This review summarises several in-depth investigations on ICG usage in colorectal surgery.

## 2. Materials and Methods

The following searches were conducted to perform an updated literature review on the topic of qualitative and quantitative ICG use in colorectal surgery: (1) a first search for the relevant terms “(qualitative OR quantitative) AND ICG AND colorectal AND surgery” on Pubmed was carried out on 16 May 2022, which yielded only a few (17) results; (2) the same terms were searched on www.sciencedirect.com, with 463 results; (3) another search was performed on www.oxfordjournals.com and returned 49 results; (4) the terms “ICG AND colorectal AND surgery” were searched on www.scopus.com, which returned 254 results; and (5) one final quest was conducted on www.reaxys.com for the terms “ICG AND colorectal”, which yielded 459 documents. The searches were initially performed during May 2022 and updated on 9 November 2022. Another complementary search was performed on 23 January 2023 on www.scopus.com using the terms “(the stability of ICG) OR (the photosensitivity of ICG). The 213 results were further sorted using the term “colorectal” and by a date more recent than 2018, and thirty-one articles were ultimately identified. The above findings (from searches 1 to 5) were further selected by setting the language to “English”, and duplicates and results not consistent with the subject were eliminated. Furthermore, the remaining results retrieved for our topic were summarised into principal fields of ICG application in photodynamic and combination therapy and are discussed in a brief section at the beginning of the “Results” section, while the main body of the article concerns indocyanine green and its use in colorectal surgery, as indicated by the title.

Optical imaging comprises various non-invasive methods for monitoring specific physiologic, pathological, and molecular processes and the behaviour of particular cell populations, as presented by Ottobrini [1] and Pirovano [2]. These methods detect visible-to-near-infrared photons from fluorescent and bio/chemo-luminescent substances.

Before discussing ICG use in colorectal surgery, a few thoughts on cancer and ICG therapy are necessary. This first set of data discusses ICG’s photosensitivity, stability, and oncology therapeutic potential in colorectal cancer research.

Photodynamic therapy (PDT) may treat cancer without side effects (see Table 1). PDT photosensitizer indocyanine green (ICG) is the only one approved clinically, among the NIR fluorophores. Water volatility and poor singlet oxygen quantum yield prevent the therapeutic use of ICG. To improve PDT, unbound ICG molecules produced another compound through hydrophilic–hydrophobic interactions on the gas–liquid interface, as shown in the research conducted by Yang [3]. Another study on the stability of ICG was reported by Lee [4], in which liposomes and phosphatidylcholine-line nanoparticles (PC-NPs) partially stabilized ICG. Z/PC-potential NPs (zein phosphatidylcholine hybrid nanoparticles) are superior drug carriers to PC-NPs. Combination treatment may enhance colorectal cancer prognoses due to tumour heterogeneity. Therefore, pH-responsive supramolecular hydrogels based on the association between bortezomib (BTZ) and, of course, indocyanine green (ICG) were produced to treat colorectal cancer using photothermal/photodynamic and chemotherapy, as presented by Qing [5]. BTZ and ICG inhibited cancer cells in vivo and in vitro. mPEG-luteolin-BTZ@ICG and laser treatment offered novel advanced colon neoplasm treatments and physiologically safe tumour therapies. Another study on the possible effects of combination therapy was reported by Ravichandran [6], who examined nonionic polysorbate-based nanoparticles that delivered piperlongumine (PL) and indocyanine green intracellularly for combined chemo/photothermal/photodynamic cancer treatment (ICG). Cancer cells selectively apoptozed due to PL-induced oxidative stress. A study reported by Bi [7] described a one-pot approach to manufactuing small nanoparticles of lipid-indocyanine green (ICG), (with the abbreviated form of “L-ICG NPs”) for the treatment of colorectal neoplasms. In the scientific context of a NIR-mediated cancer treatment that has been limited by photothermal drugs’ low NIR absorption and photosensitive molecules’ poor loading efficiency, Choi [8] created rGO, a nanocomposite under the form of a mesoporous silica-coated reduced graphene oxide that could encapsulate indocyanine green (ICG) and increase PTT/PDT efficiency in vivo and in vitro. ICG-encapsulated nanocomposites increase photothermal action and produce numerous tumour-toxic ROS. Thermo-sensitivity in relation to ICG stabilisation and combinatorial anticancer therapy was studied by Liu [9]. In situ-created photothermal network-based thermosensitive hydrogel (PNT-gel) from supramolecular cross-linking conjugated polymers stabilizes ICG and offers excellent combinatorial photothermal/photodynamic anticancer therapy. Ren [10] created a disulfide bond block copolymer to overcome issues regarding the properties of ICG and its therapeutic potential. The amphiphilic polymer delivered DOC (docetaxel) and ICG (indocyanine green) for photothermal chemotherapy. An NIR-guided photothermal treatment improved DOC and ICG solubility, and the drug delivery method maximised the therapeutic effect. Wu [11] associated the FDA-approved NIR dye indocyanine green (ICG) and anticancer medication doxorubicin to construct a novel near-infrared (NIR)-triggered dual-targeted nano-platform (FA/TPP-DINPs) for mitochondrial combined photothermal chemotherapy. FA/TPP-DINPs improved MCF-7 cytotoxicity and treatment efficacy as expected. The NIR-triggered dual-targeted nano-platform offers a revolutionary medication delivery method. Zhou [12] found that near-infrared light-responsive mesoporous polydopamine nanoparticles efficiently inhibited tumour cell proliferation and accelerated cell apoptosis, indicating that the natural active compounds EGCG and diallyl trisulfide have excellent synergy and can improve the anti-tumour effect of EGCG.

Regarding the interface between the therapeutic potential of ICG and its use in colorectal surgery in a study by Obinu [13], the authors found that freshly manufactured poly(ethyl 2-cyanoacrylate) nanoparticles (PECA-NPs) are cytotoxic to cancer cells, making them potential anticancer agents. Indocyanine green (ICG) also represents an unstable luminous material that clears quickly in vivo. This study charged ICG in PECA-NPs, improving its aqueous stability and cancer cell detection. ICG-loaded NPs and cancer cells interacted microscopically and ultrastructurally in vitro. Coated vesicles (<100 nm) in the cytoplasm suggested endocytosis by TEM. Thus, ICG-loaded NPs may diagnose and cure tumours.

As stated, researchers have created “all-in-one” nano-platforms for cancer imaging and PTT/PDT therapy. Their biological value is limited by their intricate architecture, challenging fabrication, carrier usage, and severe adverse effects. Huang [14] reported nano-platform (ICG-MB) self-assembly from FDA-approved dyes indocyanine green (ICG) and methylene blue (MB) without excipients for cancer fluorescence imaging and combinational PTT/PDT. Clinical evidence shows easy, effective, biocompatible nano-theranostic cancer phototherapy.

## 3. Results

ICG is a fluorescent dye, as described by Morales-Conde [15] in the 2022 guidelines for using fluorescence dyes in general surgery. It aids in visualising various anatomical features, such as the biliary tree, ureters, parathyroid glands, and thoracic duct. ICG also demonstrates the various characteristics of the blood supply to tissues (with examples including anastomosis in colorectal, oesophageal, gastric, bariatric, and plastic surgery procedures, as well as liver resection). Additional examples include strangulated hernias and intestinal ischemia, and all of the previously mentioned applications are visually explained in Figure 1a,b. The use of ICG in general surgery appears promising, but conclusion-drawing standardisation and randomised studies are required. According to Ghuman’s [16] informed opinion on the use of fluorescence in colorectal surgery, the adaptability of this technology enhances numerous colorectal procedures. The only method to see tissue perfusion, cancer implants, and vital structures such as ureters and lymph nodes was with white light. Fluorophores can be injected into biological tissues in order to identify them. The majority of intravenous fluorophores are composed of methylene blue and indocyanine green. Novel fluorophores targeting tumour markers have been investigated for cancer detection. Thammineedi [17] referred to luorescence-guided cancer surgery (FGS) as a new paradigm and centred research on its application in the context of abdominal-thoracic cancers. As illustrated in Figure 2a–g, Martinez Lopez [18] described novel FGS applications being developed to aid in the detection of peritoneal metastases and the evaluation of tumour resection margins. Heeman [19] discussed the significance of a standardised protocol for fluorescence imaging and data collection in clinical trials.

Lutken [20] investigated the new quantitative indocyanine green angiography (Q-ICG) technique that offers surgeons an objective assessment of tissue perfusion. In total, 13 of 1216 studies were analysed. The maximum and relative maximum intensity did not detect anastomotic leakage. Peak time, slope, and t1/2max better matched clinical objectives. Only two trials employed intraoperative Q-ICG.

Noltes [21] developed a paradigm of ICG-angiography with standardisation and quantification (WISQ) that may be used in surgical innovation independent of the user. Before predicting and preventing postoperative organ dysfunction in a heterogenous and large surgical batch, the WISQ must be prospectively validated in a larger series. Slooter [22] conducted a thorough assessment of all techniques utilised in the intraoperative fluorescence angiography (FA) of gastrointestinal anastomoses and identified thresholds for predicting patient outcomes such as leakage and necrosis. Twenty studies assessed fluorescence time, three contrast-to-background ratios, two pixel intensities, and two numeric classification scores. Seven research studies used manually measured time, and thirteen used software-generated fluorescence–time curves. To predict patient outcomes, manually measured time and parameters of the fluorescence–time curves (such as Fmax, TR, T1/2, and slope) were cut-off values. The best FA concentration metric was time to fluorescence. Fluorescence–time curves can be used to determine numerous parameters and prevent intensity persistence. Study design, fluorescence imaging equipment calibration, and software validation must be agreed upon for future data comparisons.

A standardized technique is essential for quantitative ICG angiography parameters since they are altered by varied situations, according to Ahn [23]. Fluorescence images are optimized using ICG-specific modes at 4–5 cm. Fluorescence quantification software is available but untested. Gosvig [24] reported another work that used two software-based quantification methods (FLER and Q-ICG) to analyse quantitative ICG-FI evaluations of relative perfusion in an experimental setting. Quantitative fluorescence examinations of the two software programs showed high or poor perfusion. Clinically, these changes are uncertain. Gorspas [25] FGS has good sensitivity, contrast, and specificity without radiation. It can determine the boundaries and location of superficial lesions during surgery, enabling the early diagnosis and precise excision of minor lesions. Imaging equipment development and standardisation and clinical trials to validate FGS technology’s efficacy are expected to minimise iatrogenic injuries and increase postoperative survival and quality of life.

### 3.1. Recto-Colonic Perfusion

To objectively analyse fluorescent signal angiograms using indocyanine green (ICG) in colorectal anastomosis for cancer surgery, Gomez-Rosado [26] found parameters affecting its transient intensity and pattern. The study also investigated high- and low-risk populations’ abilities to predict anastomotic leaks (ALs) and determine cut-off values. Quantitative ICG fluorescence measurements during colorectal surgery stratifies AL risk safely. The fluorescence intensity at the resection site is highest towards the rectum due to hypertension and anastomosis. The division site is shifted when fluorescence intensity at the resection site are below 169 U or slopes below 14.4 U/s to prevent vascular AL. Another study by Mo Son [27] used indocyanine green (ICG) angiography to quantify colon perfusion patterns to discover the best predictor of anastomotic problems after laparoscopic colorectal surgery. Laparoscopic fluorescent colorectal cancer imaging is a video analysis and modelling program generated perfusion graphs from colonic flow fluorescence intensity after gently injecting 0.25 mg/kg ICG into peripheral blood vessels. T1/2MAX, fluorescence slope, and time ratio (TR = T1/2MAX/TMAX) classified colon perfusion as rapid, moderate, or slow. Clinical and quantitative perfusion parameters predicted anastomotic issues. Quantitative ICG perfusion analysis using T1/2MAX and TR may detect regions with inadequate perfusion, reducing laparoscopic colorectal surgery anastomotic issues. Nerup [28] conducted a study to determine whether q-ICG and ICG-FA could enhance blood supply evaluations by operating surgeons with varying levels of expertise. A porcine model had 13 small intestinal segments with varied degrees of devascularization and two healthy fake segments. Students, residents, and surgeons performed q-ICG, ICG-FA, and white light (WL) tests. Q-ICG appears to help all surgeons safely resect healthy tissue, unlike traditional WL. The conclusion of the research presented by Tang [29] was that ICG might substantially reduce AL incidence in colorectal surgery without increasing the operative duration or postoperative complications, a concept also underlined by Vargas [30]. A meta-analysis by Trastulli [31] found that ICG-FA administered during colorectal surgery significantly reduced anastomotic leaking and surgical reintervention, particularly in patients with low or ultralow rectal resections. Similarly, sigmoid and rectal resections can have an enhanced AL (lower fistula rates) due to ICG-NIFA technology, as presented by Neddermeyer [32].

Low anastomoses and preoperative radiotherapy are significant leakage risk factors.

Many surgeons conduct unneeded protective ileostomies, which raise expenses and require reintervention. In a study research by Brescia [33], indocyanine green was used to determine intestinal perfusion in tissue treated with neoadjuvant radiation with a low anastomosis. Indocyanine green may reduce ileostomy in high-risk situations by monitoring colorectal anastomose perfusion. The lack of a standard evaluation is the biggest issue. The conclusion of the research by Meijer [34] was that similar intraoperative fluorescence results might result in different surgical strategies, demonstrating the difficulty of interpreting uncorrected fluorescence signals. The quantification and standardisation of real-time NIR fluorescence perfusion imaging may aid surgeons. Quantitative assessments of indocyanine green fluorescence imaging may help to prevent anastomotic problems during robot-assisted sphincter-saving surgeries, especially in patients with a short descending mesocolon and a high ASA status, according to Cheon Kim [35]. A study by Ishii [36] indicated that ICG fluorescence angiography may minimise AL in laparoscopic rectal cancer patients. Higashijima [37] examined 79 colon cancer patients who had double-stapling laparoscopic colorectal resection. Indocyanine green fluorescence duration assessed oral stump blood flow. AL instances and FT-AL links were explored. Stoma diversion or resection may avoid AL in delayed FT (>60 s or 50–60 s with three risk factors). Aiba [38] examined the clinical effect of the time elapsed to arterial perfusion (TAP) on an anastomotic leakage rate (AL), notably in patients lacking ICG demarcation, to estimate blood supply in such cases. Used methods: ICG-A assessed TAP in 110 colorectal surgery patients. ICG delineation requires modifying the transection line and measuring TAP at the new stump. Patients were categorised as marginal flow (MF) or direct flow (DF) by arterial routes. The third quartile or slower TAP was delayed for each group. Delayed TAP may indicate high-risk AL in MF without ICG demarcation. Diverting stomas or strict supervision may be appropriate.

AL prevention options include ICG angiography and transanal drainage tubes (TDTs). Based on animal models, the microbiome was recently found to play a role in AL development, but whether this process can occur in people is uncertain. A study by Kawada [39] analysed this aspect. Insufficient intestinal perfusion requiring a proximal relocation of the transection site was linked to high faeces volumes, suggesting a relationship between intestinal perfusion and postoperative diarrhoea. ICG fluorescence at the transection site was associated with TDT-measured faeces volume. Grafitsch [40] examined serosal oxygen saturation trends during colorectal resections using VLS. VLS was used on the colonic serosa and intestinal perfusion was assessed during left-sided colorectal resections.

The main outcome measure was anastomosis StO2. Rectal resection elevated colonic StO2. That AL onset and StO2 decline are unrelated reveals that AL development is complicated. Seeliger [41] measured mucosal and serosal perfusion in an ischemic colon section. Mucosal ischemia dominated serosal. These data show that serosal intestinal perfusion evaluations may underestimate ischemia. The best resection margin and anastomotic location need research.

In rectal cancer surgery, high and low IMA ligation (HL and LL) continue to be debated for perfusion and anastomosis leakage. Han [42] investigated this aspect using ICG. Rectosigmoid or rectal cancer patients were randomly assigned to high- or low-risk (LL) groups. HSL video image analysis was used to analyse ROI values following IMA ligation and ICG administration. T max and Slope max differed between groups, while F max did not. Anastomosis leaking, neoadjuvant chemoradiation, and F max correlated significantly. T max rose and slope max fell considerably in the HL group after IMA ligation. In contrast, IMA ligation did not influence perfusion intensity (F max). Munechika [43] demonstrated the safety and viability of high IMA ligation for descending colon cancer without compromising the distal colon by evaluating blood flow in the residual colon under fluorescence.

AL may be the hardest anterior resection complication. ICG was given intravenously and rectal stump blood flow was measured laparoscopically before a double-stapling anastomosis. Iwamoto [44] evaluated AL and numerous variables. AL patients had a longer T0 (*p* = 0.03) between intravenous ICG delivery and ICG removal from the injection channel. No additional significant differences were found between AL and non-AL groups. AL patients had longer T0s. Only suitable patients can obtain DS if prolonged T0 is identified intraoperatively.

Lower rectal cancer with lateral lymph node metastases is local in Japan and recommended LLND. Laparoscopic operations are difficult. To preserve them, remove the autonomic nerve and ureter from the vesical-hypogastric fascia. Multiple branching patterns make lymph node dissection near the internal iliac artery challenging. Ryu [45] and Gi-la-Bohorquez [46] tested ICG-based ureter and vascular navigation with a near-infrared fluorescence ureteral catheter (NIRFUC) with indocyanine green. The study found that laparoscopic LLND can use fluorescence navigation of blood arteries and the ureter to increase patient safety. Figure 3a,b show luminous ureteral intraoperative devices for better anatomical identification and dissection.

Kim [47] used fluorescence to assess the learning process for robotic full mesorectal excision with lateral pelvic node dissection in locally advanced rectal cancer patients.

The subjective visual assessment (SVA) of colour, pulsations, and bleeding from cut edges does not predict conduit perfusion after colon pull-up surgery for corrosive oesophageal strictures. Kumar [48] found mild hypoperfusion inadequacies, and ICG-FI may enhance conduit perfusion visual assessments. Spagnolo [49] examined fluorescence imaging in intraoperative intestinal assessments during gynaecological surgery. All bowel evaluation studies using indocyanine green (ICG) in gynaecology or endometriosis surgery were reviewed. Ianieri [50] noted that ICG monitoring intestinal vascularisation may reduce anastomotic leakage and rec-to-vaginal fistula, making it useful for endometriosis surgery and bowel evaluations in gynaecological cancer therapies.

Real-time hypovascular characterisation of endometriotic nodules shows bigger nodules and a lower microvessel density, helping surgeons pick the best transecting line and procedure.

Laparoscopic and intrarectal ICG angiography measures intestinal perfusion and mucosa vascularisation. Raimondo [51] found that ICG fluorescence can guide intraoperative decision-making and rectal shaving after intestinal anastomosis and discoid resection, reducing anastomotic leaking and recto-vaginal fistula in low anterior resections.

### 3.2. Improving the Detection of Cancer Tissue and Targeted Treatment

Endoscopic imaging is the principal tool for identifying gastrointestinal disorders that are harmful to human health, and a number of advancements and new techniques have been reported by Li [52].

The white light endoscopy (WLE) method was described as the first endoscopic inspection technique and remained the primary step in gastrointestinal disease diagnoses. Histological diagnosis is unreliable for gastrointestinal diseases. Recently, innovative endoscopic approaches have improved endoscope detection precision. Chromoendoscopy (CE) uses biocompatible dyes to contrast normal and diseased tissues. Narrow band imaging (NBI) uses filters to visualise vascular anatomy and improve the contrast between capillary vessels and arteries in the submucosa. FICE, the colour enhancement via flexible spectral imaging method, uses reflectance spectrum approximation to capture individual spectral images and three chosen spectral images to improve mucosal surface images. I-Scan technology captures images and enhances contrast using postprocessing techniques.

Full cytoreduction is the best long-term prognostic factor for colorectal peritoneal metastasis patients.

Subclinical peritoneal implants can be found using indocyanine green fluorescence imaging. Nonetheless, quantitative fluorescence analysis as a method has not yet been fully standardised. Gonzales-Abos [53] examined the sensitivity and specificity of quantitative indocyanine green fluorescence-q-ICG in detecting non-mucinous colorectal peritoneal metastases. Intravenous indocyanine green was administered 12 h before surgery. White light and indocyanine green nodules were identified for cytoreduction. Fluorescence was measured ex vivo. Histology revealed 37 (71.1%) malignant nodules from 52 removed. Five (13.5%) particles were fluorescence-only. In total, 8 (53.3%) of 15 non-cancerous nodules showed negative fluorescence under white light. With a sensitivity of 90.0 and specificity of 85.0%, fluorescence of more than 181 units may indicate cancer, while uptake of fewer than 100 units indicates benign disease. Quantitative indocyanine green may help evaluate non-mucinous colorectal peritoneal metastases. Fluorescence uptake above 181 units indicates malignancy, while below 100 units indicates benign illness. The technology collects photographs and enhances contrast via postprocessing.

Full cytoreduction is the best long-term prognostic factor for colorectal peritoneal metastasis patients. Fluorescence imaging with indocyanine green appears to be beneficial for finding subclinical peritoneal implants in these individuals. Similar findings were reported by Moran [54]. Lieto [55] observed that intraoperative ICG-FI had considerably better diagnostic performance than preoperative (*p* = 0.027) and intraoperative (*p* = 0.042) conventional methods, suggesting that it may improve patient outcomes following CS for PC due to CRC. Lwin [56] found fluorescence-associated molecular indicators in preclinical and clinical studies. According to Mieog [57], Privitera [58], Nagaya [59], and Tipirneni [60], several fluorescently tagged antibodies, peptides, particles, and other cancer signature molecules have been created to illuminate the target lesion. Zhang [61] discusses current breakthroughs in activable molecular probes for fluorescence-guided surgery, endoscopy, and tissue biopsy.

New methods are being developed to bring these imaging agents to clinical practice, even if few have gone past early-stage trials.

To provide patients with a real-time intraoperative visualisation, target selection, imaging agents, and detecting systems must work together.

A precise and effective tumour inhibition method, metal-organic framework-mediated chemo-photothermal therapy guided by photoacoustic imaging (PAI) synergistically induces immunogenic cell death and increases immune cell infiltration into the tumour microenvironment, making it more sensitive to an immune checkpoint blockade (aPD-L1).

This treatment may reduce the systemic toxicity of colorectal cancer drugs and enhance immune checkpoint blockades. Liu [62] loaded oxaliplatin (OXA) and indocyanine green (ICG) into HA-modified MIL-100 (Fe) nanoparticles to create multifunctional nanoparticles. For imaging-guided treatment, OIMH NPs showed sensitive photoacoustic ageing (PAI) and synergistic effects when chemotherapy was combined with photothermal therapy (PTT) to eliminate tumour cells. Chemo-photothermal therapy may sensitise cells to the immune checkpoint blockade response (aPD-L1), causing systemic anti-tumour immunity through immunogenic cell death (ICD) and T-cell activation. The observed tumour suppression was produced by chemotherapy, PTT, and aPD-L1.

### 3.3. Lymph Nodes: Mapping and Metastasis

Indocyanine green is widely used to map sentinel lymph nodes in various cancers safely and easily.

Indocyanine green has not been widely used due to conflicting research results. Thus, Villegas-Tovar [63] conducted a meta-analysis to evaluate indocyanine green in mapping sentinel lymph nodes and metastases in colorectal cancer patients undergoing surgery. A comprehensive search for relevant English and Spanish research was conducted without time constraints.

For the meta-analysis, HSROCs and random effects models synthesised quantitative data. Specificity, sensitivity, and positive and negative probability ratios were determined for each HSROC curve. Laparoscopic indocyanine green sentinel lymph node mapping is more accurate for colon cancer, research shows, and poorly detects lymph node metastases. A study by Li [64] describes a technique for fluorescently staining excised lymph nodes using paired-agent molecular imaging. The control agent’s signal accounts for any nonspecific retention of the targeted agent.

Continuous dual-needle infusion of an antibody-based imaging agent pair (EGFR labelled Cetuximab: IRDye-800CW; control agent: IRDye-700DX-IgG) or Affibody-based imaging agent pair (ABY-029; control agent: IRDYN-700DX carboxylate).

In 22 min of tissue processing, a single 0.2-mm micro-metastasis in a complete lymph node could be identified with >99% sensitivity and >95% specificity.

The detection capabilities of intraoperative lymph node biopsy methods such as frozen pathology with 20% micro-metastasis sensitivity are substantially surpassed.

### 3.4. Inflammatory Bowel Disease (IBD)

Anastomotic leaks (ALs) following restorative proctocolectomy and an ileal J-pouch increase morbidity and pouch failure. J-pouch creation involves perfusion testing. ICG-NIRF lowers ALs, although its use in restorative proctocolectomy is questionable. Eder [65] sought to standardise pouch surgery ICG-NIRF and AL studies. A prospective experiment aimed to achieve an AL within 30 days following restorative proctocolectomy with an ileal J-pouch for ulcerative colitis at an IBD referral centre. For the postoperative investigation, three intraoperative ICG-NIRF perfusion visualisations were filmed. Descriptive analysis and logistic regression identified quantitative clinical and technical components (secondary outcome) and the main result. J-pouch AL has been redefined and categorised. A postoperative pouchoscopy was tested for AL. An amount of 25 patients had no AL on intraoperative ICG NIRF visualisation or postoperative visual analysis. The anastomotic site was fluorescent in all ALs (category 2). (4 of 25). Anastomosis was unchanged. The ICG-NIRF visualisation was repeatable. The visual interpretation of ICG-NIRF may not always identify pouch ALs, according to the facts that the study indicated.

Quantitative and objective interpretation may be needed in the future. A conclusion from research by Spinelli [66] was that FA might help minimise perfusion-related anastomotic leaks after IPAA surgery, and a prospective, randomised trial is necessary to validate this hypothesis.

### 3.5. Studies Quantifying Perfusion through the Association with Fluorescence and Other Methods

Surgeons visually inspect tissue to assess colonic perfusion. Fluorescence angiography provides qualitative data, but the interpretation is contentious. Fluorescence and physiological tissue characteristics such as oxygen saturation are unknown. Soares [67] examined intestinal tissue oxygen saturation and fluorescence intensity. Egi [68] measured tissue oxygen saturation (StO2) in the intestinal tract using near-infrared spectroscopy, which measures oxygen concentrations precisely and provides objective results instantly. The arrival time until ICG reaches the tissues was suggested as a possible indicator.

We hypothesise that the time it takes ICG to reach the anastomotic region following intravenous injection can be used to compare ICG fluorescence angiography to gastrointestinal tract StO2 data.

Intestinal artery occlusion causes acute mesenteric ischemia.

Treatment includes revascularisation and necrotic intestine removal. Necrotic intestines can be hard to spot in therapy. Mehdorn [69] demonstrated that hyperspectral imaging (HSI) and indocyanine green fluorescence angiography (ICGFA) may be useful for objective intraoperative intestinal perfusion monitoring.

Comparative research was performed between quantitative fluorescence angiography and hyperspectral imaging for intestinal ischemia evaluation, as proposed by Barberio [70]. Splitting arcade branches generated ischemia bowel in hyperspectral imaging and fluorescence-based AR. Hyperspectral imaging measured tissue oxygenation. Fluorescence angiography was conducted with 0.2 mg/kg intravenous indocyanine green with a near-infrared laparoscopic camera. Perfusion maps were built using proprietary algorithms from the time-to-peak fluorescence signal and layered on real-time images to create fluorescence-based augmented reality. Selecting and superimposing nine neighbouring zones of interest onto real-time images creates hyperspectral-based augmented reality. Augmented reality with fluorescence and hyperspectral permitted comparisons. Capillary lactate was measured in interest areas.

Both imaging methods produced two local capillary lactate prediction models. The study measured intestinal perfusion with hyperspectral and fluorescence angiography. Fluorescence angiography was less accurate than hyperspectral imaging. Hyperspectral imaging-based augmented reality may measure intestinal ischemia intraoperatively without contrast.

Poor arterial blood flow and venous congestion generate anastomotic issues that must be addressed together. Quero [71] tested software to recognise intestinal ischemia patterns using fluorescence signals. The conclusion was that computer-assisted dynamic fluorescence signal analysis identifies intestinal ischemia types.

Pfahl [72] quantified hyperspectral and ICG data to improve tissue perfusion measurements. First, two data processing pipelines to recreate an ICG-FA correlating signal from hyperspectral data. The results were technically compared to colorectal resection data. Reconstructed pictures matched 87% of 46 datasets. ICG-FA and HSI in one imaging system may improve tissue vascularisation, minimise perioperative mortality, and accelerate surgery. There are concurrent anastomotic problems. Jansen-Winkeln [73] found that in 115 patients with colorectal resections, intraoperative HIS (hyperspectral imaging) was safe, repeatable, and not disruptive to the surgical process. Additionally, it measures gut surface perfusion. As an intraoperative guiding tool, HSI may reduce postoperative problems.

The greatest constraint in perfusion evaluations using indocyanine green fluorescence angiography during colorectal surgery is that the surgeon subjectively assesses the quality of perfusion. The best intestine viability test is objective, minimally invasive, and reproducible. Kojima [74] examined the quantitative value and repeatability of laser speckle contrast imaging for colorectal surgical perfusion evaluations. According to the study, laser speckle contrast imaging can assess intestinal perfusion during colorectal surgery in real-time, quantitatively, and reproducibly without contrast chemicals. Jonas Hedlund Ronn [75] observed that Q-ICG and LSCI may be complementary rather than interchangeable. Angle and distance greatly affect LSCI. In contrast, q-ICG is slightly impacted by shifting experimental circumstances and is more easily used with minimally invasive techniques.

Joosten [76] investigated the use of ICG in an acute setting and reported that the in-traoperative use of FA alters surgical choices for intestine resection for intestinal ischemia, possibly permitting gut preservation in one-fourth of patients. Prospective studies are required to improve the optimal use of this technology for this indication and to establish standards for the interpretation of FA images and the potential need for second-look procedures.

Vaassen [77] developed algorithms for the automatic extraction of inflow-based parameters, including the time to reach 50% of maximal intensity (T (1/2) and the maximal normalized slope (Slp n). These values better describe clinical outcomes than intensity measurements. An automated technique generates immersive cartograms and provides objective bowel viability information after vascular repair. The aim of the research by Park [78] was to evaluate the ability of AIRAM to predict the anastomotic complication risk in patients undergoing laparoscopic colorectal cancer surgery. When the ICG graph pattern stepped up, conventional quantitative parameters lost accuracy, but AI-based categorisation maintained accuracy. The analysis improved statistical performance verification. D’Urso [79] researched fluorescence-based enhanced reality (FLER), a computer-based quantification method for evaluating bowel perfusion using fluorescence angiographies. This prospective study examined clinical feasibility and correlated FLER with metabolic perfusion markers during colorectal resections. FLER visualises the quantified fluorescence signal in augmented reality and estimates bowel performance reproducibly. The goal of a study by Tokunaga [80] was to objectively measure intestinal temperature with the aid of thermography, which can be helpful for assessing blood perfusion. Thermography is a new way to quantify intestinal blood perfusion. The temperature boundary is visible in thermographic photographs. The residual intestinal tract temperature was substantially higher than the resected intestinal tract at the targeted separation line.

### 3.6. Estimation of Optimal Resection in Relation to Liver Function for Colorectal Metastases

ICG’s primary use in visceral surgery is to evaluate the perfusion of gastrointestinal anastomoses and to promote lymph node dissection. During staging laparoscopies or cytoreductive surgery, it may potentially be utilised to identify liver metastases, as shown by Knospe [81].

Radiological and quantitative liver function assessments have been used for hepatic surgery patients. Current surgical procedures, mainly in the setting of cirrhosis/fibrosis or after chemotherapy, are becoming increasingly complicated and lengthy. An extra examination may be needed before such surgeries. The goal of a review by Morris-Stiff [82] was to characterise the present level of knowledge on pre- and postoperative quantitative assessments of liver function in patients undergoing liver resection (hepatectomy) and liver transplantation. The review found that indocyanine green clearance is the most common dynamic hepatic function measurement. Large hepatectomy on people with low hepatic function is difficult. Surgery may be contraindicated if hepatic failure is likely afterward. In a case report by Kato [83], a right hepatectomy effectively treated the first patient with reduced liver function (ICGR15: 28%), which allowed the preservation of the complete caudate lobe. ICGR15 and ICGK were 21% and 0.12%, respectively, although all metastatic lesions decreased after treatment. Right portal vein embolism (PVE) was indicated by 39% liver volume. Portography showed considerable preservation of the right caudate lobe branch (PV1R) from the right branch of the portal vein. Liver function was re-assessed after 18 days. ICGR15 (21–28 percent) and ICGK (0.12–0.10 percent) rates decreased. Patients with marginal liver function could safely undergo the surgery. A study by Li [84] examined the connection between preoperative liver function indicators and near-infrared fluorescence-guided surgery (NIRFGS) ICG fluorescence intensity. Liver damage in mice showed increased absolute fluorescence intensity. Liver-damaged model mice and normal mice cleared ICG from the tumour similarly. Background clearance was slower than in normal animals, increasing the ideal tumour-to-background ratio length. Correlation research found the strongest correlations between preoperative liver function markers and hepatic ICG clearance. Liver injury extends the optimal TBR time, increasing the surgical window. This study demonstrated that NIR fluorescence imaging delays preoperative liver function tests.

### 3.7. Flap Assessment

GMI (the interposition of the gracilis muscle) is often used to treat complicated perineal fistulas. Fistula recurrence, necrosis, and poor wound healing make operative success less probable. Intraoperative muscle flap perfusion cannot be measured. Lobbes [85] examined an innovative, objective GMI ICG-NIRF software evaluation. ICG-NIRF visualisation data from five inflammatory bowel disease patients having GMI for perineal fistula and repair was retrospectively analysed. The new programme created ROI perfusion curves for each GMI. Curve shape, maximum slope value, distribution, and range indicated satisfactory perfusion.

### 3.8. Experimental Research and Results Foreshadow Future Clinical Achievements

Decreased intestinal perfusion is believed to contribute to the pathogenesis of necrotising enterocolitis (NEC). Research by Knudsen [86] was designed to examine intestinal perfusion in NEC lesions. During laparoscopic and open surgery, quantitative fluorescence angiography using indocyanine green (q-ICG) was performed. Thirty-four caesarean-born pre-term pigs received parenteral nutrition and escalating amounts of infant formula to develop neonatal enterocolitis (NEC). Macroscopic NEC lesions were graded 1–6 during surgery. q-ICG and a computer method for pixel intensity measured intestinal perfusion. q-ICG appears to be a viable method for evaluating tissue perfusion in NEC lesions. Neurons glow near-ultraviolet light recently. Dip [87] investigated whether nerves glow brighter than background and vascular structures under NUV light and at what intensity nerves are best visible. Fluorescence scores of 200 indicated 100% accuracy in identifying nerves from other anatomical structures in vivo at all NUV levels. According to Barth [88], nerve signals to background were measured in animal models.

Wang [89] synthesised a new cyclic TMTP1 homodimer, TMTP1-PEG4 ICG, which detected tumours more sensitively than its monomer. It was an intriguing photothermal chemical that reduced cancer development when exposed to NIR laser light and could distinguish lymph node metastases from normal lymph nodes. Gulfam showed dual NIR light responsiveness in anticancer therapy using very porous and injectable hydrogels made from cartilage acellularized matrix [90].

Multispectral dye analyses are being studied to increase accuracy. Polom [91] employed a single-camera system for two near-infrared wavelengths and two fluorophores—ICG and MB—during colorectal surgery. The study revealed that two fluorophores might be employed in colorectal surgery without interfering with quantitative analysis. Two dyes in one treatment may increase fluorescence for different purposes. Image-guided surgery’s optical technology will let fluorophores see different structures. Van Beurden [92] noted that multiwavelength fluorescence guiding, which integrates complementary fluorescent readouts during surgery, would offer future problems.

Although laparoscopic excision for early gastric and colorectal cancers is becoming increasingly prevalent, loss of touch sensibility makes stomach and intestinal tumour detection challenging. Lee [93] suggested using ICG-loaded alginate hydrogel as a luminous surgical marker for accurate laparoscopic procedures. The best ICG-HSA complex concentration was 30 M, and a 1:1 mole ratio of HSA and ICG generated the maximum fluorescence intensity. After 3 h, mice subcutaneously injected with ICG or ICG-HSA solution diffused the fluorescence signal around the injection site. Twenty-four hours later, faint fluorescence was detected. ICG-HSA-loaded alginate gel prevented injection-site dispersion and prolonged fluorescence detection to 96 h. Laparoscopic surgery identified porcine stomach hydrogel injection sites after three days. This laparoscopic alginate hydrogel surgical marker is accurate and durable. Marston [94] sought an optical imaging agent for FGS technology in CRC. The authors compared panitumumab-IRDye800CW to isotype control IgG. Three CRC cell lines (LS174T, Colo205, and SW948) were transplanted into mice and imaged using open- and closed-field fluorescence. Tumour-background fluorescence ratios were calculated to quantify fluorescent contrast. After 10 days, mice were killed and their tumours stained for microscopic analysis. Pani-tumumab-IRDye800CW produced a greater fluorescence contrast than IgG-IRDye800CW in a mouse model of colorectal cancer, making it suitable for FGS.

Optical imaging (OI) provides real-time clinical imaging and genetic, morphological, and functional illness data.

Kan [95] developed a novel interventional OI method that allows the in vivo visualisation of three pathologic zones of ablated tumour periphery to quickly identify residual tumours during radiofrequency ablation (RFA) therapy. The eight rabbits with orthotopic hepatic tumours were separated into partial and complete RFA groups. Interventional OI using indocyanine green distinguished ablated tumour, transition margin, and residual tumour or surrounding the normal liver. Quantitative signal-to-background ratio comparisons were made between the three zones and incompletely and completely ablated tumours. The association between ex vivo OI and pathology validated interventional OI. Interventional OI might identify residual tumours by distinguishing partly and completely ablated tumour margins. This technique may provide new ways to evaluate tumour elimination in a single interventional ablation session.

Spatial and temporal coordination of brain neuropeptides occurs under stress. Peptide release, transport, and breakdown in the brain must be studied to understand their stress response roles.

Over the past two decades, genetically encoded fluorescent calcium indicators have greatly improved our understanding of how particular neuronal activity regulates behavioural and physiological responses to stress.

In addition, several structural features of neuropeptide GPCRs have been uncovered. Welch [96] and Padurariu [97] revealed that enteric neurons and enterocytes produce OT and have developmentally regulated OT receptors, which is linked to gut neural growth (OTRs). The findings suggest that OTR-mediated signalling regulates enteric neuronal activity, mucosal homeostasis, intestinal permeability, and intestinal inflammation.

Bergenheim [98] and Raducanu [99] revealed that intestinal stem cells at Lieberkühn crypt bases generate children that replace resident cells lost from the villi tip during homeostatic processes.

Organoids can be generated in vitro, and orthotopic transplantation in mouse models of mucosal injury showed that intestinal organoids spontaneously connected and integrated into the injured epithelium, accelerating healing and weight gain.

This suggests that intestinal stem cell transplantation in humans may actively promote mucosal healing and be utilised in the treatment of various gastrointestinal disorders, including inflammatory bowel disease, where mucosal healing is a key treatment goal and the best predictor of clinical remission.

In the preclinical era, monitoring transplanted cells in vivo is essential to assess engraftment and wound healing.

The feasibility of labelling intestinal organoids with fluorescent dyes and nanoparticles for confocal laser endomicroscopy (CLE) was explored. CLE evaluated organoid homogeneity, durability, cell viability, differentiation, and efficiency. CLE and fluorescent dye-based tagging can follow intestinal organoids after transplantation to verify implantation.

Any antibody or tiny cancer-targeting molecule can be bioluminescent. Fluorescence and bioluminescence imaging (FI/BLI) quantify tumour volume, evaluate experimental drug targeting, and distinguish cancer therapy’s main and secondary effects in preclinical research. Visualising cancer-specific fluorescent probes are being developed to stage cancers, monitor novel therapies, and improve surgical resection and image-guided biopsies. According to research by Woo [100], (1) fluorescent proteins that are physiologically safe, stable, and clearly visible with a high target-to-background ratio and (2) extremely sensitive optical detectors are crucial FI components.

The images in Figure 1, Figure 2 and Figure 3 and their subsets have been reproduced in this article courtesy of KARL STORZ SE & Co. KG^©^, Germany.

Present research accomplishments have been reported by De Galitiis [101] in p53 gene mutations in colorectal cancer using fluorescence-assisted mismatch analysis (FAMA). Thorough research on p53, gene polymorphism in a specified population might be conducted using FAMA by focusing on a specific location (based on a geodemographic classification, such as Romanians and their genetic polymorphism, as described previously by Murarasu [102,103]). Because the discovery of more frequent mutations might indicate earlier detection and treatment, such an intervention may consistently contribute to cancer control in that region.

In addition, q-ICG could be utilized to diagnose and treat a localised infection in the abdominal area, such as either an intraperitoneal (Xie [104]) or retroperitoneal site (Marincas [105]).

## 4. Discussion

Several non-invasive optical imaging methods monitor physiologic, pathologic, and molecular processes and cell populations. Indocyanine green, a luminescent product, qualitatively identifies anatomical features (the biliary tract, ureters, parathyroid gland, or the thoracic duct) and estimates tissue blood flow.

There is a narrow window of opportunity for optimal visual conditions, necessitating extensive preoperative planning, including agent administration well in advance of the planned operation. This method also discards potentially valuable information from the initial phase of the dynamic process. The field of ICG fluorescence angiography is ripe for the application of computer visualisation to more precisely and objectively distinguish the dye’s presence and course, given that the fluorophore creates a specific contrast in tissue that is easier to discern against the background of normal tissue.

Different systems or software solutions provide ICG fluorescence quantification; however, intraoperative applications are uncommon and quantification techniques vary widely. Although the evidence is promising, ICG use in general surgery needs standardisation and larger randomised studies to draw some final findings.

Intraluminal evaluation of anastomotic perfusion using ICG-FA has the potential to be an innovative and fruitful application of ICG technology, as anastomotic leak (AL) is an extremely dangerous complication of colorectal surgery.

As shown by Gherghe [106] metastatic lesions in disseminated forms of cancers can be differentiated from degenerative lesions with the help of quantitative analysis using mainly SPECT-CT, but the use of quantitative methods of ICG could represent a good and non-irradiating alternative for this purpose.

The correlation between the: (a) peculiarities of lymphatic drainage, (b) the importance of the location of sentinel lymph nodes, and (c) the clinical impact of variations in lymph node distribution and flow, as shown by Voinea [107], Sandru [108], and Saha [109], suggests that the more detailed the lymphatic drainage in cancer is, then the better the therapeutic approach becomes.

As a result of a paucity of high-level evidence, colorectal surgeons have yet to employ it routinely, as some doubt its true benefits. In addition, some have raised the issue of the added expense of ICG without a scientifically demonstrated benefit. According to a recent cost analysis by Garoufalia [110] on the routine use of ICG imaging for anastomotic perfusion assessments, it is cost effective. The authors demonstrated that routine use of ICG for anastomotic assessments can reduce the financial burden of AL based on the assumptions that the cost of AL exceeds USD 5616.29 and the cost of ICG-FA is less than USD 634.44, and that the odds ratio for AL reduction with ICG-use (based on a literature review) is 0.46.

There is little question that each of these advances is theoretically realisable; nevertheless, the extent to which they can be used and how well they can be utilised in clinical contexts are the primary factors that will determine their therapeutic utility. Because these are screen-based techniques and call for an increased level of comprehension to visualise the surgical area, there is a significant demand for the sharing and growth of expertise, such as through the recording of surgical procedures on film, as suggested by Hardy [111]. Validating innovations in this way allows them to be paired with new computational methodologies such as AI in order to obtain better degrees of interpretation accuracy. In order to more successfully deploy these technologies, collaborative surgical groups need to build data sources that are independent, representative, and accessible, and they also need to collaborate closely with partners from both industry and academia.

Vasovagal or allergic symptoms, such as anaphylactic shock, hypotension, tachycardia, dyspnea, or urticaria, are extremely uncommon when ICG is administered intravenously, which makes it safe. These last two qualities, the cost-efficiency and the safety, added to all the other practical elements discussed previously, lead to the use of ICG in colorectal surgery being highly recommended.

## 5. Conclusions

NIR-assisted fluorescence surgery is new.

Dye-NIR imaging increases the visible spectrum. Thus, it may provide more precise anatomical and functional information, enabling complete tumour excision and preservation of vital normal tissues. Table 2 illustrates the optimum timing for ICG before fluorescence imaging and it consists of a chronological display of the data presented in Dip’s work [112], a consensus among 140 intercontinental experts.

In colorectal surgery, indocyanine green fluorescence is used quantitatively and qualitatively.

Multiple studies have demonstrated that intraoperative fluorescence imaging may safely and effectively evaluate anastomotic perfusion during colorectal surgery and reduce leaks, improving patient outcomes. New uses for indocyanine green, including colorectal liver metastases detection and management, are emerging.

These advances may help surgeons and patients by improving general surgery and surgical oncology.

## Figures and Tables

**Figure 1 medicina-59-01530-f001:**
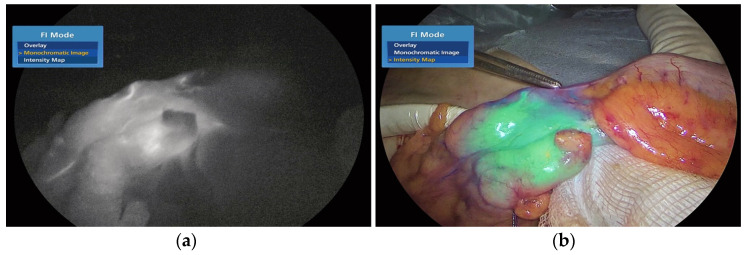
a and 1.b. ICG-enhanced fluorescence-guided evaluation of colonic perfusion following division of the mesentery during laparoscopic left colectomy. Intraoperative Monochromatic NIR/ICG (**a**) and Intensity Map (**b**) images. The forceps tip marks the well-perfused area.

**Figure 2 medicina-59-01530-f002:**
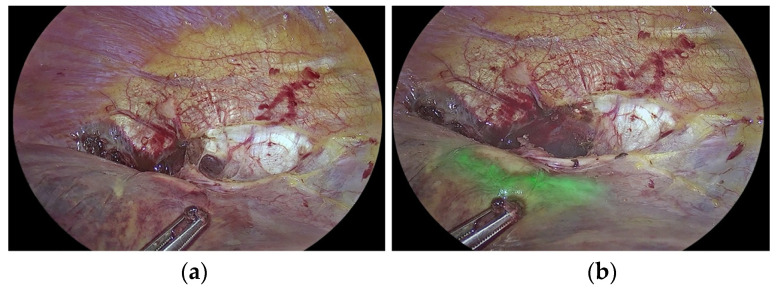
In this situation, ICG-enhanced fluorescence imaging helps locate metastatic tumours and determine resection margins. Figure 2 Enhanced fluorescence-guided liver resection for metastatic hepatic lesions (with ICG). The intraoperative images (**a**–**g**) were captured using the following visualisation modes: (**a**) White light mode, (**b**) NIR/ICG Overlay mode, (**c**) NIR/ICG Intensity Map mode, (**d**) NIR/ICG Overlay mode (open surgery), (**e**) NIR/ICG Overlay mode (Macroscopic image of surgical specimen following extraction), (**f**) NIR/ICG Monochromatic mode of same specimen as in (**e**), and (**g**) NIR/ICG Intensity Map mode showing the same specimen as in (**e**).

**Figure 3 medicina-59-01530-f003:**
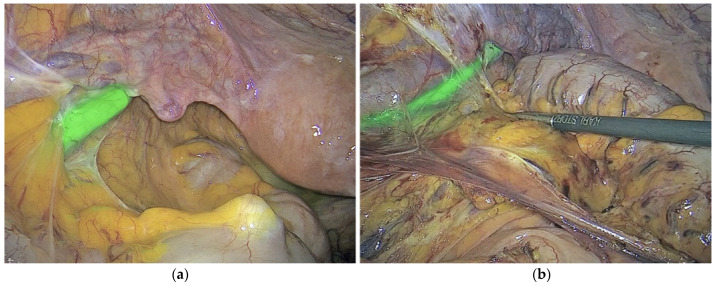
Left ureter fluorescence imaging during left colectomy with ICG: (**a**,**b**) intraoperative pictures from green NIR/ICG Overlay mode.

**Table 1 medicina-59-01530-t001:** The bioavailability of ICG and various examples of systems/substances that may influence its effect, in the context of photodynamic therapy.

Name of the Compound	Mechanism	Property/Properties
ICG-NBs-O_2_	ICG + free O_2_ nanobubbles	Exhibited better aqueous solution stability compared with free ICG. Assessed in the PDT of cancer.
L-ICG NPs 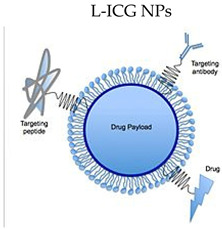	ICG + nanoparticles	ICG’s concentration-dependent aggregation, short half-life, poor photostability, low hydrolytic stability, non-specific protein binding, and non-specific targeting limit its theranostic use in cancer therapy. ICG in NP platforms may solve these issues.
Phosphatidylcholine-line nanoparticles (PC-NPs) and zein particles 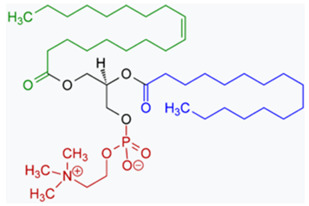 1-Oleoyl-2-palmitoyl-phosphatidylcholine	ICG + PC-NP or ICG + Z PC-NPs	It has been partially effective to stabilise ICG by encapsulating it in liposomes, phosphatidylcholine nanoparticles (PC-NP). The Z/PC-NP also inhibited ICG degradation more effectively than the PC-NP.
Bortezomib 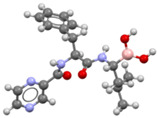	ICG + Bortezomib	Bortezomib is known as an anti-neoplasm drug involved in the treatment of multiple myeloma and mantle cell lymphoma, among others
Piperlongumine 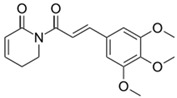	ICG + Piperlongumine	Piperlongumine (also known as piplartine or piperlongumin) is an amide alkaloid found in the fruit of the long pepper (Piper longum) plant, which is naturally found in southeast Asia. After the excision of tumours, an alginate containing piperlongumine will be used to neutralise any remaining cancer cells. The hydrogel demonstrated efficacy in laboratory and animal investigations, and Phase I human clinical trials are programmed to start to begin in 2023 (according to the web page: https://en.wikipedia.org/wiki/Piperlongumine, accessed on 01 July 2023)

**Table 2 medicina-59-01530-t002:** Optimum timing of ICG before fluorescence imaging, arranged chronologically and displayed in association with the main use of ICG in each of the particular surgical field mentioned. Part of the data from the table has been retrieved from the work presented previously by Dip, as mentioned in bibliographic [112].

The Field	The Time	The Main Applications
Plastic surgery	20–60 s	ICG angiography in the assessment of free flaps, pedicled flaps, or large skin paddles
Colorectal surgery	30–60 s	Perfusion evaluation, intraoperative ureteral visualisation, sentinel node detection, lymphatic drainage visualisation.
Lymphedema surgery, thyroid and parathyroid surgery	<1 min	Indocyanine green (ICG) lymphography is used for lymphovenous anastomosis (LVA) imaging. NIR ICG fluorescence imaging is promising for real-time parathyroid gland location during thyroid surgery.
Gastric surgery	11–30 min	ICG fluorescence angiography determines the primary blood supply to the proximal stomach before dissection during sleeve gastrectomy to prevent damaging/injury of these arteries.
Laparoscopic surgery	>30 min	Laparoscopic surgery introduced ICG fluorescence imaging to increase anatomy visualisation and to decrease the probability of injury to vital structures during dissections.

## Data Availability

Not applicable.

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
