# Peer review of "Indocyanine Green (ICG) and Colorectal Surgery: A Literature Review on Qualitative and Quantitative Methods of Usage"

_medicina, 2023, doi:10.3390/medicina59091530_

Round 1

Reviewer 1 Report

1. I am not sure what the aim of this systematic review was. You mention it is about "qualitative and quantitative ICG use in colorectal surgery" but is this a term you coined? If this term has been reported before please cite that publication. 

2. It is confusing that the results section starts with definitions of optical imaging and then has an exhausting description of technical things such as Z/PC-NP etc. The results should be alligned with the aim of the review; for example, I am not sure how Z/PC-NP is related to the "qualitative and quantitative ICG use in colorectal surgery". 

3. please explain all abbreviations you use the first time you use them. For example, PDT on line 75.

4. Have you registered this systematic review?

5. lines 193-194: not sure why this sentence is a separate paragraph. Please explain. 

6. If this is a systematic review, you need to present a flow chart of study collection. 

7. Line 384: "these people" is a phrase that cannot be included in a scientific paper. 

8. Discussion section lacks depth and is too short.

9. Please use the PRISMA guidelines. 

Editing is needed.

Author Response

AUTHORS’ ANSWERS TO REVIEWER’S 1 COMMENTS AND OBSERVATIONS

First of all, thank you for your comments! Secondly, we would like to mention  that we greatly appreciate the opportunity we have been given to improve on our work with the help of the indications and the requests provided/asked  by both   reviewers. We sincerely hope that we were able to successfully make the modifications asked. We created a table in which we tried to address every comment point-by-point.

 Thank you very much!

Respectfully yours, Dr Ionescu and team.

Reviewer 1’s Comments

Author’s answers

1. I am not sure what the aim of this systematic review was. You mention it is about "qualitative and quantitative ICG use in colorectal surgery" but is this a term you coined? If this term has been reported before please cite that publication. 

1.    The article is not a systematic review, but a narrative one(a so-called “simple” review). No, I have not coined either of the two terms. As far as I have noticed while documenting the article, ever since the discovery of ICG and of its use in colorectal surgery, various articles emerged initially on the qualitative methods of use of ICG, methods which mainly referred to its optic or fluorescent effect.  It was only  after different measurements were applied  to the method in question . when   the terms “quantitative use of ICG in colorectal surgery” were employed. In this latter category of methods, parameters were used for calculations that aimed to  clearly objectify the results and make them non-subjective.

2. It is confusing that the results section starts with definitions of optical imaging and then has an exhausting description of technical things such as Z/PC-NP etc. The results should be alligned with the aim of the review; for example, I am not sure how Z/PC-NP is related to the "qualitative and quantitative ICG use in colorectal surgery". 

2.    I have moved the general information on optical image and various technical things regarding the photodynamic therapy in the introduction section. All the technicalities in this section were abridged. Photodynamic therapy that uses ICG  as a photosensitizer can be applied also in colorectal cancer treatment (and can be also administered intraoperatively  and therefore I have considered it related to colorectal surgery). Z/PC-NP and PC-Np are different types of drug carriers. If you consider that this section or information does not benefit the article or is not suitable here at all, then of course that  I can and I will erase it.

3. please explain all abbreviations you use the first time you use them. For example, PDT on line 75.

3.    Done

4. Have you registered this systematic review?

4.    It is not a systematic review, but a narrative one.

5. lines 193-194: not sure why this sentence is a separate paragraph. Please explain. 

5.    Yes, indeed the sentence belongs to the previous paragraph and I have added it to that place.  

6. If this is a systematic review, you need to present a flow chart of study collection. 

6.    It is not a systematic review

7. Line 384: "these people" is a phrase that cannot be included in a scientific paper. 

7.    I have rephrased it.

8. Discussion section lacks depth and is too short.

8.    I have expanded the “Discussion” section.

9.   Please use the PRISMA guidelines. 

9.    It is not a systematic review.

10.                 Extensive editing of English language required

10.I have done some extensive editing through all the manuscript; if more editing is needed, I have nothing against resorting to the professional editing services of the Mdpi/ Medicina platform.

Reviewer 2 Report

The manuscript is a full-length review article, interesting to read but required revision to improve the quality.

1.       The author discusses how the ICG is linked to other nano molecules/augments the cytotoxic drug bioavailability in cancer. There is a lot of evidence in the manuscript, provided in the form of a table, to create interest for the reader.

2.       If possible, provide the ICG dye concentration which is used for the patients and bioavailability in various organs with a table.

3.        The manuscript has good information but it’s disorganized.

4.       Why the figures are disorganized? Figure 2 is there first, and Figure 1 is there last.

Author Response

AUTHORS’ ANSWERS TO REVIEWER’S 2 COMMENTS AND OBSERVATIONS

First of all, thank you for your comments! Secondly, we would like to mention  that we greatly appreciate the opportunity we have been given to improve on our work, with the help of the indications and the requests provided/asked  by both   reviewers. We sincerely hope that we were able to successfully make the modifications required. We created a table in which we tried to address every comment point-by-point.

 Thank you very much!

Respectfully yours, Dr Ionescu and team.

Reviewer 2’s Comments

Author’s answers

1.       1.The author discusses how the ICG is linked to other nano molecules/augments the cytotoxic drug bioavailability in cancer. There is a lot of evidence in the manuscript, provided in the form of a table, to create interest for the reader.

1.     As suggested, I have created Table 1 about the bioavailability of ICG in the context of Photodynamic therapy.

      2.       If possible, provide the ICG dye concentration which is used for the patients and bioavailability in various organs with a table.

2.     I have created Table 2 with the optimum timing of ICG before fluorescence imaging, which I considered that , indirectly, is correlated with the bioavailability of the substance in different organs.

      3.        The manuscript has good information but it’s disorganized.

3.     I have re-structured the first part of the “Results” section, and I have moved it to the “Introduction” section. I have rephrased extensively the entire article to make it more readable.

4.     4. Why the figures are disorganized? Figure 2 is there first, and Figure 1 is there last.

4.     It was a mistake which I hope I have now corrected( due to my editing the images “skipped” from one page to another and it is in this way that Figure 2 was accidentally displayed before Figure 1.

Round 2

Reviewer 1 Report

The authors have revised according to my suggestions.

Moderate editing is needed.